# Precision Medicine in Bladder Cancer: Present Challenges and Future Directions

**DOI:** 10.3390/jpm13050756

**Published:** 2023-04-28

**Authors:** Sambit K. Mohanty, Anandi Lobo, Sourav K. Mishra, Liang Cheng

**Affiliations:** 1Department of Pathology and Laboratory Medicine, Advanced Medical Research Institute and CORE Diagnostics, Gurgaon 122016, India; sambit04@gmail.com; 2Department of Pathology and Laboratory Medicine, Kapoor Center for Pathology and Urology, Raipur 490042, India; lobo.anandi@gmail.com; 3Department of Medical Oncology, All India Institute of Medical Sciences, Bhubaneswar 750017, India; drskmishra1984@gmail.com; 4Department of Pathology and Laboratory Medicine, Brown University Warren Alpert Medical School, Lifespan Academic Medical Center, and the Legorreta Cancer Center at Brown University, 593 Eddy Street, APC 12-105, Providence, RI 02903, USA

**Keywords:** precision medicine, bladder cancer, molecular biomarkers, histologic subtypes/variants, heterogeneity, targeted therapy, immune checkpoint inhibitors

## Abstract

Bladder cancer (BC) is characterized by significant histopathologic and molecular heterogeneity. The discovery of molecular pathways and knowledge of cellular mechanisms have grown exponentially and may allow for better disease classification, prognostication, and development of novel and more efficacious noninvasive detection and surveillance strategies, as well as selection of therapeutic targets, which can be used in BC, particularly in a neoadjuvant or adjuvant setting. This article outlines recent advances in the molecular pathology of BC with a better understanding and deeper focus on the development and deployment of promising biomarkers and therapeutic avenues that may soon make a transition into the domain of precision medicine and clinical management for patients with BC.

## 1. Introduction

Bladder cancer (BC) is the tenth most commonly diagnosed cancer with an age-standardized incidence rate (per 100,000 person/years) of 9.5 in men and 2.4 in women; globally, the age-standardized mortality rate (per 100,000 person/years) is 3.3 for men and 0.86 for women [1,2,3,4]. It is a major cause of cancer-related morbidity and mortality. According to the GLOBOCAN 2020 data, 573,278 new cases and 212,536 deaths of BC are added each year [5]. It typically affects patients in the fifth to seventh decade with a fourfold higher incidence among males [5]. BC exhibits significant morphological and molecular heterogeneity. However, despite its highly characterized molecular signature and high rate of potentially actionable genomic alterations, there has been limited success in various promising biomarker therapies [6]. Thus, this highlights the importance of a detailed and systematic validation of the various promising therapeutic modalities in precision medicine available for BC. This may soon translate to clinical management, with efforts to review the ongoing work in this area, address the obstacles in the advancement, and highlight potential solutions for implementation in clinical practice.

## 2. Pathological Staging and Histological Grading Systems

BC has been traditionally divided into non-muscle-invasive BC (NMIBC) and muscle-invasive BC (MIBC). NMIBC includes tumors confined to the mucosa and invading the lamina propria; classified as noninvasive papillary carcinoma (pTa), carcinoma in situ (CIS, pTis),or carcinomas invading the subepithelial connective tissue (pT1). A significant proportion (75%) of BC patients represent this category, characterized by frequent tumor recurrence, limited tumor progression, and higher survival rate with a lower cancer-specific mortality [2,7]. The second category of MIBC represents high-grade tumors, either locally advanced invading the muscularis propria (pT_2_), or invading the perivesical soft tissue (pT_3_), and extravesical tumors involving the adjacent organs or pelvic wall or abdominal wall (pT_4_) or metastatic tumors (pM_1_) [7]. MIBC typically receives neoadjuvant chemotherapy (NACT) followed by radical cystectomy, with a need for biomarker-guided immune checkpoint inhibitors (ICIs), as well as targeted and conjugate therapies in the locally advanced and metastatic setting.

BC represents a morphologically and genomically heterogeneous disease with a wide spectrum of subtype histologies and associated molecular alterations. The classical urothelial carcinoma (UC) is the most common type, but a diversity of morphological appearances can be displayed. The WHO fifth edition reclassified the histologic subtypes of UC as follows: infiltrating urothelial carcinoma with divergent differentiation; nested, including large nested; microcystic; micropapillary; lymphoepithelioma-like; plasmacytoid/signet ring cell/diffuse; sarcomatoid; giant cell; poorly differentiated; lipid-rich; clear cell [8]. Recognizing each subtype is critical as each carries a unique prognostic or therapeutic implication [9,10]. Recent technological advances have increased our knowledge on the genomic landscape of UC and have enhanced our understanding of the molecular features associated with the disease, as well as its subtypes. Molecular analysis of UC and its subtypes has revealed intratumoral and intertumoral heterogeneity at the genomic and cellular levels. Subtype histology are extreme examples of tumor heterogeneity, each with distinct molecular characterization. This heterogeneity ultimately translates to a variable choice and response to therapy, drug resistance, and relapse rate. As heterogeneity occurs on multiple levels, addressing these diverse alterations is crucial for clinical drug trials in order to enable appropriate targeted therapy. The discovery of molecular pathways may allow for better disease classification, prognostication, and development and deployment of novel noninvasive detection and surveillance approaches, as well as the selection of efficacious therapeutic targets.

## 3. Molecular Pathogenesis

NMIBC and MIBC display two distinct clinicopathologic and molecular phenotypes with reference to their prognosis and biologic behavior, as well as supporting evidence for two divergent pathways in their pathogenesis (Figure 1). The former is thought to originate from a benign hyperplastic urothelium, with only a minor proportion of 10% cases progressing to high-grade noninvasive and subsequently invasive UC [11]. The primary genetic alterations, known to consistently be associated with this group of tumors, include alterations in the receptor-associated tyrosine kinases for *FGFR3*, *HRAS*, *KDM6A*, *KMT2D*, and *PIK3CA.* Chromosome 9 deletion occurs in the early phase of BC tumorigenesis. *FGFR3/HRAS* mutations are most frequently noted during the development of hyperplasia and low-grade (Ta) carcinoma [11,12,13,14,15,16]. *FGFR3* is one of the most common genetic alterations, and *FGFR*-targeted *therapy* has become a promising treatment strategy in BC; *FGFR1-4* alterations in UC patients respond well to *FGFR* inhibitors such as erdafitinib and rogaratinib, and similar therapeutic results have been achieved targeting the *PI3K* pathway [17,18]. Activating mutations in the *RAS* gene activate the mitogen-activated protein kinase (*MAPK*) and PIK3 pathways. Moreover, exclusive activating mutations noted in upstream *FGFR3* and *RAS* suggested the possibility of a common downstream pathway in the carcinogenesis of BC [11]. Furthermore, the coinciding occurrence of *PIK3CA* and *FGFR3* mutations suggests a potential synergetic oncogenic effect for *PIK3CA* mutations [19]. The most common genetic alterations in MIBC include *TP53*, *KMT2D*, *KDM6A*, and *RB1* [16]. Most MIBC originates from a dysplastic urothelium, evolving into a flat CIS and finally into a high-grade noninvasive UC (pTa) through the acquisition of *CDKN2A* alterations, further progressing to pT1 carcinomas through additional genetic instability with *TP53/RB1* inactivation and finally to MIBC through the accumulation of multiple genetic and epigenetic alterations [16,20,21]. The molecular pathogenic pathway for MIBC chiefly involves alterations in the tumor suppressor genes involved in the regulation of the cell cycle, e.g., *TP53*, *P16^INK4a^*, and *RB1* [16,22,23,24] (Figure 2). The telomerase reverse transcriptase (*TERT*) promoter defect has been recently recognized to be the key driver mutation in BC, known to affect 60–80% of BC patients [25,26,27,28]. *TERT* mutations have been detected in a wide range of urothelial pathologies, including benign urothelial proliferations and tumor-like lesions, benign urothelial neoplasms, premalignant and putative precursor lesions, urothelial carcinoma and its subtypes across gender, tumor grade, and stage, and nonurothelial malignancies [25]. The *TERT* gene is located on the chromosome 5p15.33 (chr5:1,253,147-1,295,069) and encodes a subunit of telomerase with the telomerase RNA component (TERC) involved in telomere replication. Telomerase is active in gametes and cancer cells, and it helps in maintaining the length of telomeres by adding telomere repeat 50-TTAGGG-30 to the end of telomeres. Telomerase is inactivated in somatic cells; however, *TERT* promoter mutation and telomerase reactivation allow the somatic cells to bypass senescence when the telomere is critically short [25,29,30,31]. Elderly patients with BC are known to harbor a higher frequency of *TERT* mutations as compared to those younger than 50 years [25,32,33]. There are inconsistent reports of the association of *TERT* promoter mutations with stage, grade, and prognosis of patients with BC [24,25,26,27,28,29,30,31,32,33,34,35,36,37,38].

In a recent multiplatform analysis by The Cancer Genome Atlas (TCGA) bladder cancer group, using apolipoprotein B mRNA-editing enzyme, catalytic polypeptide *(APOBEC)-*mediated mutagenesis was identified as the principal mutation signature within MIBC [39,40]. The *APOBEC* family consists of seven evolutionally conserved deaminases, including activation-induced cytidine deaminase, *APOBEC1*, *APOBEC2*, and *APOBEC4*. These enzymes are responsible for DNA editing caused by the deamination of cytidines (C) to uridines (U), which are repaired to guanines (G) or thymidines (T) [40,41,42,43]. These mutational signatures are prevalent in bladder cancer, as well as cervical, breast, head and neck, and lung cancers [41,44]. There is evidence of a definite role of *APOBEC* mutations in all stages of BC, especially during the tumor progression and evolution from early stage to MIBC [40,45]. TCGA analysis revealed that *APOBEC3A* and *APOBEC3B* signatures were present in 67% of single-nucleotide variations (SNVs) among the MIBC cohort [40,41]. Although it was shown that patients in TCGA cohort with *APOBEC3*-enriched tumors showed an improved survival and better prognosis, clear-cut elucidation of this mutation as a favorable prognostic marker has not yet been established. Moreover, there is evidence that *APOBEC*-induced mutagenesis is enriched in UC treated with chemotherapy [46]. The role of *APOBEC3* enzymes in the promotion of treatment resistance in UC remains to be determined. Another mutational signature detected in approximately 20% of SNVs was associated with *ERCC2* mutations. *ERCC2* encodes a DNA helicase, known for its role in the nucleotide-excision DNA repair pathway. *ERCC2* mutations were shown to be associated with an improved response to cisplatin-based chemotherapy, as well as to immune checkpoint blockade and radiation therapy, in advanced BC. The lack of normal *ERCC2* function and somatic *ERCC2* mutational status correlated with complete response to cisplatin-based chemosensitivity in MIBC [41,47,48,49,50,51]. The third signature associated with 8% SNVs in TCGA analysis was related to 5-methylcytosine deamination [41].

## 4. Molecular Taxonomy of Urothelial Carcinoma

BCs are biologically, clinically, and pathologically heterogeneous. Multiple recent studies have identified a number of molecularly distinct gene expression clusters explaining their heterogeneity and have focused on improvising a potentially useful model based on the molecular classification of BC to prognostically stratify these tumors into relevant categories. This could be potentially useful to correlate and predict response to chemotherapy and immunotherapy, guide novel therapeutic modalities, and provide a better framework for clinical management along with enhancement of current scientific knowledge [39,40,41,52,53,54,55,56,57,58,59,60,61,62,63,64,65,66].

Over the past decade, there have been several molecular classification systems independently identified by various groups (Figure 3). These classifications are based on RNA and/or immunohistochemistry (IHC) expression characteristics [40,41,56,57,58,63,64,65,66,67]. There exists a significant overlap among all efforts to amalgamate the terminology and systems [64]. The most comprehensive working classifications include those proposed by the Lund University group, TCGA group, MD Anderson Cancer Center Group, and a simplified four-gene signature-based molecular classification with Nano Stringn Counter assay proposed by Lopez-Beltran and colleagues [39,40,52,55,56,63,65,66,67].

The earliest classification developed by the Lund University by Lindgren et al. comprised two molecular subgroups of BC: MS1, which included pTa tumors, enriched with *FGFR3* mutations (55% in MS1 vs. 7% in MS2, *p* < 0.05); MS2, which included high-grade tumors and those where *TP53* mutations were more common [64]. Subsequently, this analysis was expanded by Sjödahl et al., in their study on a larger cohort of tumors, wherein they described five distinct molecular subtypes: urobasal A type characterized by features of the normal urothelium such as keratin 5 (KRT5), P-cadherin (P-Cad), cell-cycle activity (CCNB1) restricted to the tumor–stroma interface, and *FGFR3* overexpression in the basal cells with a relatively favorable prognosis; genomically unstable type of high-grade MIBC showing proliferation throughout the tumor parenchyma and harboring *TP53*, E-cadherin (E-Cad), and *ERBB2* overexpression with absence of KRT5, P-Cad, and FGFR3 expression; squamous cell carcinoma-like (SCC-like) type illustrating a squamous cell differentiation and overexpression of basal keratins with a relatively poorer prognosis; urobasal B type sharing combined features of all three aforementioned subtypes; infiltrated type characterized by infiltration of immune cells with extracellular matrix gene expression. In the following years, this group continued to modify and refine this classification with the inclusion of two more types: small-cell/neuroendocrine-like subtype, similar to the genomically unstable type but with expression of high levels of neuroendocrine (NE) markers such as chromogranin, synaptophysin, neuron-specific enolase (NSE), and CD56; mesenchymal-like type characterized high expression of mesenchymal markers such as vimentin and ZEB2 and differed from other subtypes by showing low expression of FOXA1, GATA3, KRT5, and KRT14. A careful analysis of the molecular types and morphological classification revealed that pTa tumors were mainly of the urobasal A subtype, whereas pT1 tumors were of the urobasal A and genomically unstable subtypes, and all subtypes accounted for a certain proportion of MIBC [56,64,65,66,67].

TCGA 2017 formulated a classification for MIBC into five different categories: luminal, luminal–infiltrated, luminal–papillary, basal–squamous, and neural. All luminal subtypes highly express luminal marker genes (*KRT20*, *GATA3*, *UPK1A*, *UPK2*, *FGFR3*, *PPARG*, *FOXA1*, and *ELF3*). The luminal–papillary subtype is characterized by predominant papillary tumor morphology with a lower stage, while the luminal infiltrated subtype expresses extracellular matrix and smooth muscle genes with lymphocytic infiltrate. These tumors are also reported to have an increased expression of immune markers such asPD-L1 and PD-1. The basal–squamous subtype comprises tumors with squamous differentiation and expresses basal and stem-like markers (CD44, KRT5, KRT6A, and KRT14), squamous differentiation markers such as desmocollins (DSC1–3), desmogleins (DSG1–4), TGM1 (transglutaminase 1), and PI3 (elafin), and immune marker genes (*CXCL1* and *L1CAM*). Furthermore, this group is also enriched in *TP53* mutations and shows a female predominance and a strong immune gene signature expression, along with lymphocytic infiltrate. The neural subtype is associated with the worst clinical outcome and includes tumors with or without NE/small-cell histology, but with high expression of genes involved in neural differentiation, along with expression of NE/neural differentiation markers such as chromogranin, PEG10, PLEKHG4, and TUBB2B. Concurrent mutations in both *TP53* and *RB1*areconsidered the characteristic genetic alteration in this group of tumors [39,40,57,60,61,62,63,64,65,66,67,68,69,70,71].

The MD Anderson Cancer Center Group in 2014 devised a three-tier subtyping system with prognostic implications after analyzing the mRNA of 73 MIBC tumors and labeled them as luminal, p53-like, and basal subtypes [16,63]. The basal MIBCs were characterized by a high expression of squamous differentiation markers such as p63, were more invasive and aggressive tumors at presentation, and were associated with a dismal prognosis. The luminal group was enriched with epithelial markers, features of activated PPAR-γ, estrogen receptor, activating *FGFR3* mutations, and potential sensitivity to FGFR blockers, and they had a relatively good prognosis. The p53-like subtype shared a similar profile to the luminal subtype, along with a high frequency of *TP53* mutation. Furthermore, this group of tumors was resistant to NACT, and all drug-resistant MIBCs adopted a p53-like phenotype following chemotherapy, suggesting that the p53 gene may play an important role in chemotherapy-induced mutagenesis [16,63].

In an attempt to harmonize all the various overlapping classification systems, Zhu et al. (2020) attempted to establish a relationship and address the interrelating overlay among the various molecular subtypes [16]. They suggested that the Lund “urobasal A” subtype can be further classified into the MDA “luminal” subtypes, that the Lund “infiltrated” and MDA “p53-like” subtypes share common features of enriched extracellular matrix markers, and that the Lund “SCC-like” subtype and MDA “basal” subtype are both enriched in SCC differentiation markers.

In a most recent classification by Lopez-Beltran and colleagues (2021), three subtypes of BC were identified using a Nano String-based four-gene panel expression analysis on a series of 91 BC cases, both NMIBC and MIBC, with classical and subtype histology [26]: luminal subtype (KRT20^+^/GATA3^+^), basal subtype (KRT5^+^/KRT14^+^/GATA3^low/−^/KRT20^low/−^), and null/double negative (non-luminal/non-basal) subtype (KRT14^−^/KRT5^−^/GATA3^−^/KRT20^−^). All three categories were meaningful for overall cancer-specific survival. The luminal subtype was consistent with low aggressiveness and enriched in NMIBC, with the morphology of conventional UC, low PD-L1 expression, and low bladder cancer-related mortality. Conversely, the basal subtype was consistent with high aggressiveness, enriched in pT2–4 disease and with chiefly micropapillary, plasmacytoid, and nested subtypes. This category was also enriched in high PD-L1 expression, thus creating an opportunity for these patients to be treated with ICI [26,72,73].

## 5. Molecular Characterization of Subtype Histology of Urothelial Carcinoma and Pure Adenocarcinoma

### 5.1. Plasmacytoid Urothelial Carcinoma

The plasmacytoid UC represents an aggressive subtype of UC, composed of infiltrating plasma cell-like single discohesive tumor cells, admixed with cells containing intracytoplasmic vacuoles, resembling signet ring cells. These tumors have an overall low survival, with patients typically presenting at an advanced stage, high mortality rate, high propensity for relapse, and frequent peritoneal carcinomatosis, with some response to chemotherapy. This subtype shares IHC and molecular alterations with classical UC, such as staining for KRT7, p63, GATA3, and uroplakins, along with genetic mutations in *TP53*, *RB1*, *KMT2D*, and *ARID1A* [39,41,74,75,76,77,78]. However, the development of these tumors is additionally driven by loss-of-function mutations in *CDH1* and promoter hypermethylation of *CDH1*, which possibly also contribute to its aggressive nature [39,41,74]. These mutations are considered the defining feature specific to this histologic subtype of UC [74]. The *CDH1* loss further contributes to the higher rate of cellular migration and peritoneal spread, along with a higher incidence of local recurrence due to cell discohesion and stromal invasion, thus also contributing to a higher cancer-specific mortality [74]. Of note, in contrast to the germline *CDH1* mutations seen in diffuse hereditary gastric cancers and a subset of lobular breast cancer, no germline *CDH1* mutations were identified in plasmacytoid UC [74] (Figure 4a). 

### 5.2. Micropapillary Urothelial Carcinoma

Micropapillary UC represents another rare but aggressive subtype of UC. Many clinicians advise an early cystectomy in these tumors, even in the absence of invasion into the muscularis propria layer. Morphologically, this tumor is characterized by small tight clusters of high-grade tumor cells, lacking a true fibrovascular core, with a reverse cellular polarization and lack of cohesion between the tumor and stroma [39,41,79,80,81]. This tumor is most commonly associated with higher rates of *ERBB2* mutations, more commonly amplifications than mutations [82,83]. Higher rates of *ERBB2* amplification are observed in micropapillary UC as compared to classical UC. This amplification confers worse cancer-specific survival following radical cystectomy associated with this subtype [41,84,85]. Morphological intratumoral heterogeneity, as well as intratumoral heterogeneity of *ERBB2* amplification, is noted in tumors with mixed micropapillary and NOS UC, with*ERBB2* amplification being more common in micropapillary rather than NOS UC areas; moreover, therate of *ERBB2* amplification was higher in these mixed tumors as compared to pure NOS UC, not mixed with micropapillary component [40,86,87,88]. These findings do point toward a possible role of *ERBB2* activation in the development of the aggressive subtype of UC (Figure 4b). 

### 5.3. Small-Cell/Neuroendocrine Carcinoma of the Bladder

Small-cell carcinoma (SmCC) is a rare subtype of BC, morphologically identical to its counterpart in the lung. It can occur in its pure form, but is more usually admixed with a urothelial (invasive or noninvasive), glandular, squamous, or sarcomatous component [10]. SmCC commonly harbors combined alterations in both *TP53* and *RB1* [71,89]; however, whether this genomic instability actually favors lineage switching from oncogene-addicted urothelial cells to NE-like tumor cells, along with a decreased response to targeted therapy, is still debated. These genetic alterations have also been detected in UC that does not exhibit features of SmCC or NE differentiation. Furthermore, other alterations have also been detected in SmCC of the bladder and include *TERT* promoter mutations and truncating alterations within chromatin-remodeling genes such as *CREBBP*, *EP300*, *ARID1A*, and *KMT2D*, along with *APOBEC* somatic mutational burden and whole-genome duplication. All these events are presumed to arise early in the process of oncogenesis and reflect an evolutionary point toward small-cell lineage differentiation; however, they are unlikely to be the only transforming event, as there were multiple prior driver mutations, many of which are common in bladder urothelial cancers [39,41,71]. Moreover, the phenomenon of NE differentiation as an outcome of trans differentiation post androgen deprivation therapy, as occurs in NE carcinoma of the prostate, is not observed in SmCC of the bladder; NE differentiation in the bladder seems to develop de novo [39,41,90,91,92]. More studies are awaited to explain the association of SmCC with the neuronal/NE molecular subtype of BC, as defined by TCGA and Lund group classifications (Figure 4c). 

### 5.4. Sarcomatoid Urothelial Carcinoma

Sarcomatoid carcinoma is another extremely rare aggressive tumor subtype comprising about 0.3% of all primary urinary bladder tumors; it carries an overall dismal prognosis [10,41,93,94]. The presence of a mesenchymal component in UC is designated as sarcomatoid UC. This subtype can also exist with other subtype histologies such as glandular, squamous and/or small-cell, or NE differentiation [10,41]. The most common morphology is that of a spindle-cell proliferation; others include myxoid, pseudo-angiosarcomatous, and undifferentiated pleomorphic sarcoma-like morphology, as well as true heterologous elements such as chondrosarcoma, osteosarcoma, fibrosarcoma, leiomyosarcoma, and rhabdomyosarcoma [41,94,95]. A monoclonal cell origin has been suggested for both sarcomatous and urothelial components within the same tumor due to significant overlap in the molecular events, such as loss of heterozygosity. The clonal divergence might occur during tumor progression and differentiation [41,96]. At a molecular level, this tumor type is enriched with mutations in *TP53*, *RB1*, and *PIK3CA*, and is associated with the dysregulation of the epithelial–mesenchymal transition pathway [41,96,97] (Figure 4d).

### 5.5. Urothelial Carcinoma with Divergent Differentiation

The most common divergent histologies in UC are squamous and/or glandular differentiation [8,41]. Squamous differentiation represents the more common subtype (Figure 4e). Expression profile analysis of such tumors with divergent differentiation revealed urothelial areas as the luminal subtype and squamous areas as the basal/squamous subtype [41,98,99]. Although the association of squamous differentiation with human papilloma virus infection has been investigated, very little genomic information exists [10,41,99]. The presence of a glandular component in UC is less common than squamous differentiation [41,100]. The glandular component in UC closely resembles enteric or colonic adenocarcinoma. Molecular analysis of the glandular component revealed an increased prevalence of hotspot mutations in the *TERT* promoter region, which was not seen in other glandular lesions of the bladder, including primary adenocarcinoma of the bladder [41,101] (Figure 4f).

### 5.6. Nested Urothelial Carcinoma

Nested UC represents a rare, morphologically deceptively bland tumor, associated with an aggressive clinical course [41,102]. These tumors exhibit small, closely packed, poorly defined, and haphazardly arranged, confluent irregular nests of bland-appearing tumor cells without cytologic atypia, infiltrating the lamina propria and the muscularis propria with an associated stromal reaction [41,102]. A high rate of *TERT* promoter mutation is the only molecular finding detected in this subtype of UC to date [41,103] (Figure 4g). 

### 5.7. Adenocarcinoma

This group represents tumors characterized by a pure glandular morphology, and it also includes the entity urachal adenocarcinoma [41,104]. Morphologically, these tumors resemble colorectal adenocarcinomas. At a molecular level, these tumors are genetically distinct from UC and lack mutations in the *TERT* promoter region, as well as in chromatin-modifying genes involved in UC carcinogenesis. They are enriched in mutations in *TP53*, *KRAS*, and *SMAD4*,along with *EGFR* and *ERBB2* amplifications, thus resembling a subset of colorectal adenocarcinomas [41,105,106,107] (Figure 5).

## 6. Molecular Heterogeneity and Systemic Targeted Therapy

Tumor heterogeneity can foster tumor evolution, as well as tumor adaptation. This usually presents major challenges and can hinder personalized medicine strategies, as well as biomarker development, which depend on results from single tumor-biopsy samples [41,108,109]. An increased relapse rate has often been associated with intratumoral heterogeneity and the presence of natively resistant stem-cell populations [41,110,111]. Thus, molecular classification of MIBC and NMIBC, along with a detailed genetic analysis of the subtypes of UC, provides an opportunity for personalized medicine, biomarker development such as *FGFR3* inhibitors, and targeted therapy. The future lies in the advancement of investigational promising biomarkers and innovative trial designs, all ultimately relying on molecular subclassification. Moreover, a prognostic stratification of the various subtypes into well-defined groups remains a priority.

### 6.1. Neoadjuvant Chemotherapy (NACT) and Chemoradiotherapy

Molecular subtyping may have a therapeutic benefit, as it is the complexity at the cellular level that would have the maximum impact on the choice of therapeutic agent. Systemic cisplatin-based chemotherapy (CT), as well as cisplatin-based NACT, is the most effective treatment in advanced UC, and immunotherapy is emerging as the most viable and effective salvage treatment option in patients of UC in cases of failure of first-line CT. Major advancements in the past decade have shed light on the various genetic classes and subtypes of UC, which might vary in response to various therapeutic modalities [41,112]. Determining the response of each tumor subtype to NACT and/or immunotherapy will be critical in deciding the most efficacious targeted therapy. Genomics can also help identify patients who are more likely to have an aggressive disease phenotype with an extravesical spread; thus, identifying these patients may provide a maximum therapeutic benefit in the administration of NACT prior to surgery [113]. Furthermore, the basal subtype molecular group of BC may show maximum benefit and improved overall survival following NACT compared with surgery alone, while the luminal subtype molecular group may have the best overall survival and lowest rate of upstaging as compared to other tumors with or without administration of systemic therapy [113,114].

### 6.2. Targeted Therapy

The main gene drivers in MIBCs are *FGFR3*, *RAS*, *PPARG*, and *TERT* promoter mutations, predominantly found in the luminal subtype of bladder tumors [40]. The response to therapies targeting *FGFR3* mutations depends on intratumoral heterogeneity. Targeting *PPARG* in UC cell lines showed that inverse agonism of *PPARG* reduced proliferation rates of *PPARG*-mutant cells but not *PPARG* wildtype cells, thus pointing toward another strategy inwhich patients with luminal tumors might gain additional benefit from targeted therapy following chemotherapy [115].

### 6.3. Immunotherapy

Response to immunotherapy is dependent upon the intra- and peritumoral T-cell infiltration in response to neoantigen expression on tumor cells. Mutation load is an important biomarker to assess response to immunotherapy in patients of advanced UC [116].

### 6.4. Clinical Trial Considerations

With the expansion of our understanding about the molecular heterogeneity among the subtypes of UC, biomarkers are more likely to have a promising role in future clinical trials. The reliability of both prognostic and predictive biomarkers in the setting of tumor heterogeneity is being currently investigated. Therapeutic and clinical management decisions are made depending on the presence or absence of a particular biomarker, and the accuracy of the biomarker in predicting therapeutic response is significantly dependent on tumor heterogeneity.

## 7. Precision Medicine in Bladder Cancer

### 7.1. Biomarker

A biomarker is a measured substance or variable whose presence is indicative of or a surrogate for a disease outcome. The potential roles of a biomarker in MIBC include (1) identifying high-risk patients, e.g., patients planned for radical cystectomy ± neo adjuvant therapy, (2) predicting resistance to chemotherapy/immunotherapy, and (3) identifying pathways involved in targeted therapy. A biomarker may be prognostic (in that it provides information about the patient’s overall cancer outcome, regardless of the therapy), predictive (in that it provides information about the effect of a therapeutic intervention and, hence, can be a target), or both. An ideal biomarker is one which is reproducible, accurate, validated in multiple datasets, and most importantly, easy to use.

### 7.2. Biomarkers for Advanced Urothelial Cancers

#### 7.2.1. Biomarkers for Response to Chemotherapy (Table 1)

Cisplatin-based chemotherapy (MVAC: methotrexate, vinblastine, doxorubicin, and cisplatin; GC: gemcitabine, cisplatin/carboplatin) is the treatment of choice in patients with metastatic UC of the bladder. The overall response rates (ORRs) range from 60% to 70%, overall survival (OS) ranges from 14 to 15 months, and 5year OS ranges from 13% to 15% [117]. In patients who relapse after platinum-based chemotherapy, ORRs range from 5% to 29% with a median OS of 6.9 months (based on clinical trials of second-line chemotherapy with paclitaxel and vinflunine) [118]. In the neoadjuvant and adjuvant settings in UC, similar regimens are used to those in the metastatic setting. Most data on chemotherapy biomarkers are available for MIBC, since a pathological complete response (pCR) to platinum-based chemotherapy is prognostic in this setting.

**Table 1 jpm-13-00756-t001:** Biomarkers for response to chemotherapy.

Molecular Target	Study [Ref.]	Results	Comments
DDR Genes			
**NER pathway** ***ERCC1* expression levels**	**Bellmunt et al. [119]**	Reduced levels of *ERCC 1* mRNA expression were associated with improved survival to cisplatin-based chemotherapy in mUC.	DDR genes are not validated biomarkers for response to chemotherapy (not routinely used in clinical practice). Clinical trials are evaluating the role of PARP inhibitors in DDR gene mutated UC [126].
	**Urun et al. [120]**	*ERCC1* positivity was associated with poor survival in mUC treated with cisplatin-based chemotherapy.	
***ERCC2* mutations**	**Van Allen et al. [50],** **Liu et al. [121]**	ERCC2 mutations were associated with pCR and improved OS to neoadjuvant cisplatin-based chemotherapy in MIBC.	
	**Kim et al. [122]**	*ERCC2*-associated mutation signature single-base substitution 5 (SBS5) was associated with improved responses in mUC.	
**HRR pathway** ***BRCA* mutations**	**Taber et al. [123]**	*BRCA2* mutations were associated with SBS5 signature and responses to platinum-based chemotherapy in MIBC	
***RAD51*** **mutations**	**Mullane et al. [124]**	High nuclear staining for *RAD51*was associated with poor outcome (worse OS) for mUC patients treated with cisplatin-based chemotherapy.	
**Other DDR genes** ***ATM/RB1/FANCC* mutations**	**Plimack et al. [125]**	*ATM/RB1/FANCC* mutations were associated with improved pathologic responses and survival in MIBC treated with neoadjuvant platinum-based chemotherapy.	
***HER2/ERBB2* alterations**	**Groenendijk et al. [127]**	*HER2* missense mutations (not amplifications) were associated with response to neoadjuvant chemotherapy with platinum in MIBC.	
**Molecular subtypes of bladder cancer**	**Kamoun et al. [128]**	None of the subtypes were found to be associated with neoadjuvant chemotherapy response.	
	**Choi et al. [63]**	The p53-like subtype was chemo resistant.	
	**McConkey et al. [129]**	The basal subtype was associated with the most optimal OS in the trial of neoadjuvant chemotherapy MVAC with bevacizumab.	
	**Taber et al. [123]**	The basal/squamous consensus subtype was associated with reduced neoadjuvant chemotherapy response.	

#### 7.2.2. Cisplatin Eligibility

Cisplatin ineligibility is defined as an Eastern Cooperative Oncology Group performance status > 2, neuropathy/hearing loss grade ≥ 2, creatinine clearance < 60 mL/min, and New York Heart Association heart failure grade ≥ 3 [130,131]. Treatment with cisplatin may be prognostic in metastatic UC (mUC). Patients eligible for cisplatin and treated with cisplatin-based chemotherapy had an improved OS as compared to eligible patients not treated with cisplatin [132]. Thus, treatment with cisplatin in eligible patients (cisplatin utilization) rather than cisplatin eligibility may be a clinical biomarker for improved OS and is of paramount importance, as, even in eligible patients, around one in four is not exposed to this chemotherapy. Cisplatin-ineligible patients treated with carboplatin have a better outcome as compared to non-platinum-based chemotherapy. Lastly, the receipt of any chemotherapy leads to improved survival in comparison to no receipt of chemotherapy [132]. These data are primarily based on retrospective analysis and, hence, should be interpreted with caution.

#### 7.2.3. DNA Damage Repair Genes (DDR Genes)

Platinum-based chemotherapy leads to DNA damage through the formation of adducts and ultimately apoptosis. In a normal cell, in response to DNA damage, the DDR pathway is activated to repair the damage. This pathway comprises the nucleotide excision repair (NER) for single-stranded DNA damage, the homologous recombination repair (HRR) for double stranded DNA damage, and the Fanconi anemia pathway. Most importantly, mutations in the DDR genes lead to increased susceptibility of cancer cells to platinum-based therapy [133].

#### 7.2.4. NER Pathway

The excision repair cross-complementation group 1 (ERCC1) protein heterodimerizes with ERCC4 to form an endonuclease complex. This participates in the excision of the damaged DNA. Lower *ERCC1* levels (mRNA expression or IHC) are correlated with cisplatin sensitivity in MIBC and mUC, with improved outcomes in these patients [119]. Conversely, ERCC1 overexpression is associated with worse OS in mUC [120]. ERCC 2 mutations are associated with pCR and improved OS to neoadjuvant cisplatin-based chemotherapy in MIBC [50,121]. *ERCC2*-associated mutation signature single-base substitution 5 (SBS5) is associated with improved responses in mUC [122]. In another study on neoadjuvant GC chemotherapy in MIBC, alterations within a panel of 29 DDR genes were correlated with chemotherapy response. Deleterious DDR gene alterations include nonsense, frameshift, and splice site alterations or*ERCC2*missense mutations. The positive predictive value of a somatic deleterious DDR gene alteration for response was 89%, and the 2 year relapse-free survival was higher in patients whose tumors had a deleterious DDR gene alteration [134].

#### 7.2.5. HRR Pathway

HRR is a DNA repair mechanism, involved in the repair of double-stranded breaks and interstrand crosslinks. The undamaged homologous chromosome serves as a template for the repair of the damaged strand. *BRCA1* and *BRCA2* are prototypes for HRR genes, known for their roles as cancer predisposition genes and as predictive biomarkers for sensitivity to poly ADP-ribose polymerase (PARP) inhibitors and platinum-based chemotherapy [135]. Somatic *BRCA1/2* alterations were present in 19% of MIBC samples in TCGA, and germline *BRCA1/2* variants were observed in 2–4% of UC patients [40,136,137]. In a recent multi-omics analysis of 300 patients with MIBC or mUC, *BRCA2* mutations were associated with the SBS5 mutation signature and with chemotherapy response [135].

#### 7.2.6. ATM Serine/Threonine Kinase, Retinoblastoma Transcriptional Corepressor 1, or FA Complementation Group C (ATM/RB1/FANCC) Mutations

While *ATM* and *RB1* are cell-cycle regulators in response to DNA damage, *FANCC* is critical in interstrand crosslink repair [135]. Among other DDR genes, *ATM/RB1* mutations are considered as biomarkers of poor prognosis in unselected UC patients and may correlate with higher mutational load. *ATM/RB1/FANCC* mutations are associated with *p* < T2 response to NAC and improved OS in MIBC [125,138]. An association was also observed between high mutation burden and deleterious DDR genes. However, DDR alterations have no prognostic impact in the absence of NAC [127].

#### 7.2.7. Other Alterations

*ERBB2* (erb-b2 receptor tyrosine kinase 2) missense mutations were associated with response to platinum-based neoadjuvant therapy in MIBC [50,127]. Some other studies found no benefit of *HER2* alterations and response to chemotherapy [125]. Therefore, further analysis is required to determine the above association.

#### 7.2.8. Molecular Classifications

A gene expression profile (GEP) for a tumor is derived from the extraction and quantification of tumor RNA. GEP may open up avenues for response assessment for antitumoral therapy at the molecular level. On the basis of similarities in the GEP, clustering algorithms may be used to group tumors into molecular subtypes [135]. Accordingly, six consensus molecular subtypes for MIBC have been suggested: basal/squamous, luminal papillary, luminal unstable, luminal non specified, stroma-rich, and neuroendocrine-like; according to their similarity to basal and luminal breast cancer subtypes [40]. The utility of molecular subtypes as predictive biomarkers of chemotherapy response is unclear, and studies have produced conflicting results. The p53-like subtype included under the stroma-rich consensus subtype has been reported as chemoresistant in UC.The basal/squamous consensus subtype has been suggested to be chemoresistant in others. The basal-type tumors were shown to be the most chemosensitive in some studies [63,129]. Lastly, none of the consensus subtypes were found to associate with NAC response in the study by Kamoun et al. [128]. Co-expression extrapolation (COXEN) is a gene expression-based predictive biomarker analysis that identifies gene expression signatures in cancer cell lines associated with in vitro chemotherapy sensitivity and extrapolates those signatures to predict chemosensitivity in vivo. This model has not been shown to predict response to platinum-based chemotherapy in UC [139,140].

### 7.3. Biomarkers for Response to Immunotherapy in UC (Table 2)

In the front-line setting, ICI monotherapy has demonstrated activity in cisplatin-ineligible patients [50,130]. The choice between ICIs and carboplatin chemotherapy in this setting is not straightforward. For patients who are platinum-ineligible, ICI monotherapy is a reasonable option. However, for those who are eligible for carboplatin-based chemotherapy, a maintenance ICI approach (per JAVELIN Bladder 100) may be favored over upfront ICI monotherapy, given its proven OS benefit, as described below [141]. PD-L1 expression is used as a biomarker among cisplatin-ineligible patients to choose between ICI monotherapy and carboplatin-based chemotherapy. For patients with PD-L1^low^ tumors, upfront ICI monotherapy may be deleterious according to data on early mortality in the IMvigor130 and KEYNOTE-361 trials [142,143]. Cisplatin-ineligible patients with PD-L1-positive tumors may be considered for upfront ICI or chemotherapy followed by maintenance ICI; these options have not been directly compared in clinical trials. Currently, the strongest evidence for ICI benefit in mUC is in the post-platinum-based chemotherapy setting. Two randomized trials—KEYNOTE-045 and JAVELIN Bladder 100—have demonstrated OS benefits for single-agent ICIs as either second-line or maintenance therapy after platinum chemotherapy [141,144]. Both trials met their primary endpoint of OS in biomarker-unselected, all-comer population. Notably, the use of ICI at progression was permitted in the control arm of JAVELIN Bladder 100, and around one-third of the patients received it [145]. On the basis of these data, a strategy using maintenance ICI is preferred after platinum-based chemotherapy rather than ICI at progression. This is also the preferred approach per National Comprehensive Cancer Network (NCCN) guidelines, although there exists a risk of over-treatment in some patients.

**Table 2 jpm-13-00756-t002:** Biomarkers for response to immunotherapy.

Biomarker	Study [Ref.]	Results	Comments
**PD-L1**	**IMvigor 130 [142]**; **Keynote 361 [143]** **Rui et al.** [146]**, Litchfield et al. [147]**	Cisplatin-ineligible mUC with PD-L1^low^did not benefit from ICI monotherapy as compared to chemotherapy. Cisplatin-ineligible patients with PD-L1-positive tumors benefited from ICI monotherapy. Meta-analyses of prospective trials showed that, overall, PD-L1 expression was associated with radiographic response to ICIs in mUC patients.	PD-L1 is a biomarker in cisplatin-ineligible patients to guide the choice of upfront ICI monotherapy vs. carboplatin chemotherapy. In this population, therapeutic choices are carboplatin-based chemotherapy followed by maintenance immunotherapy. In this population, options are upfront ICI or chemotherapy followed by maintenance ICI; these options have not been directly compared in clinical trials. PD-L1 expression is the only ICI biomarker that has been incorporated into mUC regulatory approvals and treatment guidelines. PD-L1 as a biomarker is dynamic in both space and time.
**Tumor mutational burden**	**Galsky et al. [148]** **Litchfield et al. [147]**	Exploratory analyses of prospective trials in mUC suggested that the combination of TMB and PD-L1 could more effectively distinguish ICI responders and non-responders than either biomarker alone. Clonal TMB and the APOBEC signature were among the most important features associated with response in a multivariable model predicting ICI response in bladder cancer.	Challenges in implementing TMB as a biomarker include selecting an optimal cutoff and harmonizing assays.
**Somatic alterations** **TRAF2** **CCND1 amplification** **DDR genes**	**Litchfield et al. [147]** **Litchfield et al. [147]** **Mariathasan et al. [149]****,** **Powles et al.** [150]	Loss of TRAF2 was associated with ICI response. CCND1 amplification was associated with ICI resistance. Mutations in DDR pathway genes were associated with improved outcomes in exploratory analyses of both the IMvigor210 and JAVELIN Bladder 100 trials.	DDR genes alone are probably not predictive of response to ICI.The combination of DDR gene mutation and TMB is likely to be predictive.
**Gene expression** **TGFβ response signature (F-TBRS)**	**Mariathasan et al. [149]** **Galsky et al. [151]**	In IMvigor210, both a TGFβ ligand (TGFB1) and a TGFβ receptor (TGFBR2) were associated with nonresponse and reduced OS to ICI. F-TBRS was associated with response in immune-excluded tumors. A higher F-TBRS signature was also associated with worse OS with atezolizumab in the IMvigor130 trial.	

#### 7.3.1. PD-L1

PD-L1 is expressed in 20–30% mUC patients [135,152,153]. In BC, it is both a prognostic (increased expression PD-L1 by 20–30% correlates with advanced stage and worse outcomes) and a predictive marker for response to anti-PD-1 and anti-PD-L1 therapy [154]. Meta-analyses of prospective trials showed that, overall, PD-L1 expression is associated with radiographic response to ICIs in mUC patients [146,147]. At the same time, benefits from ICIs occur, regardless of PD-L1 expression. However, even among PD-L1-positive patients, single-agent ICI response rates are low and variable across randomized trials, ranging from 20% to 40% [142,143,155,156]. Although most trials analyzed the data using a prespecified cutoff for PD-L1 on IHC, the results did not consistently show improved responses with higher PD-L1 expression, which is not a very surprising observation, given that PD-L1 assays are not uniform across clinical trials (nonuniformity in the assays or scoring). While pembrolizumab and nivolumab clinical trials used the DAKO assays, Ventana assays were used for durvalumab and atezolizumab. In the pembrolizumab and nivolumab trials, PD-L1 tumor cell staining was used, whereas the IM vigor trial used PD-L1 immune cell staining. The cutoffs for PD-L1 staining were also different. Variability in the staining platforms and cutoffs, including cell types and scoring system, may have been responsible for variability in the observed responses with different immune checkpoint inhibitors [153]. Other important factors which should be considered in using PD-L1 as a standalone biomarker to assess response to immunotherapy is the intratumoral heterogeneity with regard to PD-L1 expression and its dynamic nature in space and time during the disease course [154,157]. Attempts were made to harmonize PD-L1 assays in non-small-cell lung cancer and found consistent staining across some assays, but not with the others [158,159]. Limited inter-observer reliability in scoring PD-L1 staining on immune cells was also described [160]. The application of liquid biopsy and immune-targeting tracers for positron emission tomography (ImmunoPET) [161,162,163], may be a possible and more efficient way for serial monitoring of PD-L1 or other ICI biomarkers.

#### 7.3.2. Tumor Mutation Burden

The tumor mutational burden (TMB) is defined as the total number of mutations per coding area of a tumor genome. A higher number of mutations increase the chances of generating neo-tumor-antigens, which can be recognized by the host immune system as immunogenic neoantigens [164,165,166]. TMB is quantified as the number of coding somatic mutations per megabase (MB) of DNA [165]. Tumors with high TMB have been demonstrated to have a microenvironment rich in immune cells and associated cytokines [167]. Bladder cancer is the most highly mutated cancer [168]. TMB has been linked to ICI response in mUC [149,169,170]. Pembrolizumab is approved as a therapeutic option across solid tumors with TMB ≥10 mutations/Mb without satisfactory treatment alternatives [171]. In the IMvigor210 trial, TMB assessed by targeted genomic profiling of 315 cancer-related genes (Foundation Medicine) correlated with a longer OS and ORR with atezolizumab independent of PD-L1 expression. Patients whose tumors had the highest mutation load (≥16/MB)) had a significantly longer survival compared with patients whose tumors had lower mutational loads (<16/MB) [HR 0.37, (95% CI 0.21–0.64)] [130,172]. TMB did not correlate with PD-L1 expression; however, it may be useful as an adjunct to other biomarkers in predicting outcomes with ICIs [173]. The combination of TMB and PD-L1 may be more efficacious together than either biomarker alone in predicting response to ICI [151,174]. Mutational signatures (denoting underlying tumor mutation) attributed to the APOBEC family of cytidine deaminases are frequently seen in BCs [40,175]. These were predictive of favorable responses to ICIs in mUC [147,149,151]. Recently, a meta-analysis across multiple cancer types including mUC suggested that clonal TMB (in all cancer cells in the clone) followed by total TMB was most predictive of response to ICIs. In addition, clonal TMB and the APOBEC signature were among the most important features associated with response on multivariate analysis [147]. As with PD-L1, selecting an optimal cutoff and harmonizing assays have been the common challenges in implementing TMB as a biomarker. In addition to the quantity (cutoff), the quality of the mutations (short insertions/deletions), clonality (clonal versus sub clonal), and the association of the neo antigens with the patient’s HLA may be considered while assessing TMB as a biomarker [147].

#### 7.3.3. Molecular Subtypes of Bladder Cancer

Basal type tumor cells have higher PD-L1 expression [176]. In the IMvigor 210 trial with atezolizumab, the luminal cluster II subtype had a statistically significant higher response rate compared to luminal cluster I, basal cluster I, and basal cluster II subtypes [117]. According to the results from this study, combining the Lund molecular classification scheme with TCGA scheme could lead to better prediction of responses. Tumors that were both genomically unstable (GU) in the later classification and luminal II had high TMB and better responses to ICI. On the contrary, tumors that were luminal II but not GU had low TMB and lower responses [149]. In the CheckMate 275 trial with nivolumab, improved responses were seen in basal I subtype followed by luminal II subtypes [177]. In the JAVELIN Bladder 100 trial of maintenance avelumab, however, there was no association between the TCGA subtypes and OS [141]. There is, thus, a heterogeneity in the outcomes with immunotherapy with respect to molecular subtypes. It may be plausible that the current molecular classifications of UC may not be adequately representative of appropriate molecular signatures predictive of response to ICIs; hence, further research is needed.

#### 7.3.4. Gene Expression Profiling of the Tumor and Microenvironment

Tumor immunity is the result of a complex interaction between the tumor cells and immune cells in the tumor microenvironment (TME). A comprehensive immune gene expression profiling of these cell types, along with their chemokine and cytokine repertoire, may represent the ongoing interactions resulting in tumor immunity. As gene expression profiling is a dynamic display of ongoing cellular processes in the tumor and cells in the TME, it is more reflective of the molecular pathways involved at the time of sampling [135]. Two broad categories of gene expression signatures have been linked to ICI response in prospective mUC cohorts: a group of genes reflecting cytotoxic T-cell activity associated with ICI response; a group of genes reflecting immunosuppressive stromal signaling associated with ICI resistance. These signatures remain exploratory pending validation in additional prospective cohorts [135]. A variety of inflammatory gene signatures reflecting CD8^+^T-cell activity and/or interferon-gamma signaling have been associated with ICI response in mUC. Some recurrent genes in these signatures include *CCL5*, *CD27*, *CD8A*, *CXCL9*, *CXCL10*, *CXCR6*, *GZMA*, *GZMB*, *IDO1*, *IFNG*, *LAG3*, *PRF1*, *STAT1*, and *TBX21* [135]. In the IMvigor 210 trial in mUC, a higher CD8^+^T effector signature (PD-L1 positivity on the immune cells was associated with the expression of genes in a CD8^+^T effector set) correlated with higher complete response rates to atezolizumab. Similarly, CXCL-9 and CXCL-10 (chemokines representative of the T effector signature) expression had a higher response to immunotherapy [149]. Notably, CXCL9 expression was one of the strongest predictors of ICI response in the Litchfield et al. meta-analysis of ICI biomarkers across tumor types [135]. In the Checkmate 275 study, a higher value of 25-gene interferon-gamma (IFN-γ) signature was associated with a higher response to nivolumab [177]. While IFN-γ is known to have favorable effects on antitumor immunity, persistent signaling has been associated with adaptive resistance to checkpoint therapy. One of the most important IFN-γ mediated effects is the increased expression of PD-L1 and PD-L2 [178]. Prolonged exposure of cancer cells to IFN-γ signaling leads to expression of a number of ligands for T-cell inhibition, which in turn leads to resistance to ICIs independent of the PD-1/PD-L1-pathway [179]. An eight-gene subset of that signature focused on CD8 T effector activity was positively associated with response in IMvigor210 [149]. TGF-β signaling in the tumor stroma creates an immunosuppressive phenotype or immune-excluded phenotype in that the cytotoxic T cells are separated from the tumor cells by a dense fibrous stroma, promoting angiogenesis and metastases. On the basis of data from the IMvigor210 study, Mariathasan et al. showed that increased pan fibroblast TGF-β response signature (F-TBRS), TGF-β ligand (TGFB1), and a TGF-β receptor (TGFBR2) in fibroblasts within the peritumoral stroma were associated with a lack of response and poorer survival to atezolizumab, especially in patients where CD8^+^ T cells were excluded from the tumor parenchyma [149]. A higher F-TBRS signature was also associated with worse OS for patients treated with atezolizumab rather than platinum chemotherapy in the IMvigor130 trial [151].

### 7.4. Biomarkers for Targeted Therapy in UC

The basis of targeted therapy is the specificity of treatment directed against a target that is preferentially altered in the cancer cells as compared to the normal cells. Three such targeted therapies have been approved in mUC (Table 3 and Table 4),although many other targets have been evaluated but not yet approved. The targeted therapeutic molecules approved in UC include *FGFR* inhibitor erdafitinib, Trop 2 inhibitor sacituzumab govitecan, and Nectin-4 inhibitor enfortumumab vedotin (EV). Each of these molecules is approved in the second-line setting after progression on first-line platinum/non-platinum-based chemotherapy. As of now, only erdafitnib is recommended on the basis of the *FGFR* alteration status (mutation/fusion); thus, *FGFR* alterations serve as a biomarker for benefit from *FGFR* inhibitors. The other two molecules are approved irrespective of the biomarker results. There are no head-to-head trials comparing second-line chemotherapy, immunotherapy, and targeted therapy. Historically, responses with second-line chemotherapy have been dismal at around 10% with a median survival of 7–9 months [118,180,181]. Even in the second line, the response rates with immunotherapy are to the tune of 13–20% with amedian OS of 10 months [144,155,177,182,183]. In that context, most of these targeted therapies have been approved on the basis of their superior response rates as compared to the historical results, although the overall survival also compares favorably with immunotherapy. Erdafitinib, a pan-FGFR-kinase (FGFR 1–4) inhibitor, was approved in 2019 for patients with locally advanced or mUC with progression after platinum-based chemotherapy with known susceptible *FGFR2/3* alterations. The specific alterations include *FGFR3* mutations or *FGFR2/3* gene fusions. The approval was based on a phase II trial demonstrating overall response rates of 30–40% in a biomarker-driven population [17,184]. EV consists of a monoclonal antibody specific for Nectin-4 conjugated to monomethyl auristatin E (MMAE), a microtubule-disrupting agent [185,186]. Approval was also granted for locally advanced or mUC after prior platinum-based chemotherapy and ICI as a result of the phase II EV-201 trial. In the confirmatory phase III EV-301 trial, EV conferred a significant survival benefit over standard chemotherapy in the post-chemo/post-ICI setting, leading to regular FDA approval [187]. Notably, EV has shown benefit and is approved for treatment without regard to Nectin-4 levels. Considering recent data supporting a maintenance strategy with avelumab after platinum-based chemotherapy for advanced UC, as well as results from the EV301 study, EV may be a reasonable option at the time of the first relapse after maintenance immunotherapy [141]. EV in combination with pembrolizumab has been accorded a breakthrough therapy designation as first-line treatment for metastatic disease on the basis of a higher response rate and duration of response [188,189]. Regimens containing EV are being evaluated in the first-line (ClinicalTrials.gov numbersNCT04223856andNCT03288545) and perioperative (NCT03924895) settings. The third targeted therapy approved in mUC is SG (a monoclonal antibody specific for Trop-2 conjugated with SN-38), the active metabolite of irinotecan [190]. Approval for SG was recently granted in April 2021 after the phase II TROPHY-U-01 trial. This demonstrated a 27% ORR and 10.9 months median OS in the post-chemo/post-ICI setting [191]. Similar to EV, SG has been tested and approved without regard to the levels of its target, Trop-2.

Table 4 enumerates biomarkers for response to targeted therapy.

#### 7.4.1. FGFR

*FGFR* alterations are ubiquitous in UC. The most common FGFR3 alterations are mutations which account for 80% of the FGFR alterations in NMIBC and almost half of the alterations in MIBC [196,197]. FGFR alterations occur in 20% of the patients with advanced urinary bladder UC and up to 37% of the upper tract (UT) UC [24,198]. Of these, *FGFR3* alterations (mutations and fusions) are significant from a therapeutic perspective and are more common in UTUC than UBC [199,200]. Of the *FGFR3* mutations, S249C is the most common, accounting for up to half of these mutations [17,201]. *FGFR 2/3* mutations are enriched in the luminal type 1 molecular subtypes of UC, which are usually immune-excluded. These tumors show reduced T-cell infiltration, as well as low PD-L1 expression on TILs; hence, they are postulated to be resistant to immunotherapy [17]. Mutations in *FGFR*, which belongs to the family of tyrosine kinase receptors, bestow the cancer cells a clear survival advantage in that the receptor functions in a ligand-independent manner and the constitutive tyrosine kinase activity leads to incessant downstream signaling via the *RAS/MAP3K/PI3K* pathway, ultimately leading to cell proliferation [202]. *FGFR* fusions and amplifications are less common alterations in the *FGFR* pathway [203]. In the BCLC 001 trial of erdafitinib, up to three-fourths of the alterations were *FGFR2/3* mutations, and the remaining were *FGFR2/3* fusions [17]. As with driver mutations in lung cancer, it has been postulated that immunotherapy may not be an appropriate option for *FGFR* mutated UC. Data show that, in the immune-excluded luminal type 1 UC, limited responses are seen with immunotherapy [177,204]. In this subset enriched with *FGFR* alterations, durable responses have been seen with *FGFR* inhibitors after progression on immunotherapy [17,205]. On the contrary, the pivotal second-line immunotherapy trials in UC have shown responses irrespective of the *FGFR* alteration status, thus putting to question the notion that *FGFR* alterations are a biomarker for lack of response to immunotherapy [206]. A longer median duration of response (68% of patients with a response for at least 12 months) with fewer toxic effects of grade 3 or more (15% vs. 46%) suggests that immunotherapy may provide a better safety and efficacy profile than *FGFR* targeted therapy [144]. Thus, the optimal sequencing of therapy in *FGFR*-mutated mUC is debatable. We may expect an answer about the optimal sequencing from the phase III THOR trial that has a 1:1:1 randomization into three arms with chemotherapy, immunotherapy, and erdafitinib in *FGFR-*mutated mUC [ClinicalTrials.gov numberNCT03390504]. Lastly, although *FGFR* mutations are biomarkers for response to erdafitinib, the same cannot be said for *FGFR* fusions/alterations. *FGFR* mRNA expression and ctDNA levels of *FGFR* have also been evaluated as biomarkers for response to FGFR inhibitors. The bottom line is that *FGFR* somatic mutations are a better predictor of response to *FGFR* inhibitors compared to other alterations [197]. There is a fair degree of concordance between ctDNA and somatic *FGFR* alterations. According to the results from the BISCAY trial, baseline increased levels of ctDNA are predictive of worse OS, and serial ctDNA can be used for response assessment [195]. Resistance to *FGFR* TKIs has been observed most commonly due to mutations in the ATP-binding pocket of the *FGFR*. Both primary and secondary resistance have been observed. K650E gatekeeper mutations demonstrate primary resistance to infigratinib [207,208]. Other mechanisms of FGFR TKI resistance include activation of alternate *RAS/MAPK* and *PI3K/AKT* pathways, lysosome mediated TKI sequestration, activating gene fusions/ and epithelial-to-mesenchymal transition [207].

#### 7.4.2. Nectin-4

Nectin-4 is a cell-adhesion molecule that is highly expressed in UC and may contribute to tumor-cell growth and proliferation [185,209]. EV consists of a monoclonal antibody specific for Nectin-4 conjugated to monomethyl auristatin E (MMAE), a microtubule-disrupting agent. The delivery of the microtubule payload into the tumor cells leads to cell-cycle arrest and apoptosis [185,186]. According to the phase III randomized controlled trial, EV significantly prolonged survival as compared with standard chemotherapy in patients with locally advanced or mUC who previously received platinum-based treatment and a PD-1 or PD-L1 inhibitor [187]. However, Nectin-4 expression was not mandatory for enrolment as high expression was observed in a vast majority of patients with advanced UC [192,209].

#### 7.4.3. Trop 2

Erdafitinib is limited to patients with *FGFR2/3* mutation or fusion [210]. Many patients will still need newer therapies. Trophoblast cell-surface antigen 2 (Trop-2) is a transmembrane cell surface glycoprotein that is expressed extensively on most carcinoma cells and plays an important role in cell transformation and proliferation [211,212]. Thus, increased expression is associated with poor outcome, including mUC [190]. Sacituzumab govitecan (SG) is a Trop-2-directed molecule composed of an anti-Trop-2 humanized IgG monoclonal antibody coupled to SN-38, the active metabolite of the topoisomerase 1 inhibitor irinotecan with a high drug-to-antibody ratio (7.6 molecules of SN-38 per antibody) [213]. Internalization of Trop-2-bound SG delivers SN-38 inside tumor cells, thus killing the tumor cells, whereas the hydrolyzable linker enables SN-38 to be released into the tumor microenvironment, killing adjacent cells (bystander effect) [214,215]. Approval for SG was recently granted in April 2021 after the phase II TROPHY-U-01 trial, and it demonstrated a 27% ORR and 10.9 months median OS in the post-chemo/post-ICI setting [191]. Similar to EV, SG has been tested and approved. A benefit with SG was even observed in a small subgroup with prior exposure to EV. Responses in patients previously treated with EV suggest various nonoverlapping mechanisms of action and resistance between the two antibody–drug conjugates [190]. The results from this phase II trial will be corroborated in the ongoing phase III confirmatory trial of SG versus taxane or vinflunine in mUC (TROPiCS-04; ClinicalTrials.govidentifier NCT04527991). Additional cohorts of TROPHY-U-01 continued to evaluate the role of SG in mUC. Cohort 2 is investigating the role of SG in platinum-ineligible patients with mUC who progressed after immunotherapy. Cohort 3 is evaluating SG in combination with pembrolizumab in patients with mUC who progressed after prior platinum-based chemotherapy and are immunotherapy-naïve. Both cohorts 4 and 5 are evaluating SG as induction and maintenance therapy in mUC patients who responded to induction platinum-based chemotherapy in the neoadjuvant setting, comprising chemotherapyeither alone (cohort 4) or in combination with avelumab (cohort 5) [190].

#### 7.4.4. Other Targeted Therapies

*HER2* is amplified in a subset of patients with UC. *HER2* amplification is an adverse prognostic event in UC. Yet, anti-HER2 therapy has not been proven to improve outcomes in mUC. One of the reasons is the heterogeneity in HER2 testing in trials with anti-HER2 therapy. In a retrospective analysis performed by our group, comparing HER2 IHC and *HER2* FISH results demonstrated that ASCO/CAP HER2 testing guidelines for breast cancer could be implemented in UC [153]. Hence, patients with true amplification of HER2 can be evaluated in future clinical trials utilizing anti-HER2 therapy in UC.PARP inhibitors have not been found to improve outcomes in *BRCA*-mutated/*HRR*-deficient UC [194]. The incidence of *BRCA1/2* mutations is about 1.5% and 1.4%, respectively, in UC [136]. Previously, it has been mentioned that defects in the DDR pathway may predict responses to chemotherapy. An innovative approach could be the combination of chemotherapy and PARP inhibitors in DDR-deficient UC. Other therapies that target *PI3K/AKT/mTOR*, *MAPK*, and *VEGF* pathways have also been investigated in UC. The RANGE study found that ramucirumab, a monoclonal VEGFR-2 antibody with docetaxel, improved the progression-free survival, but not OS in previously treated mUC patients [216].

## 8. Conclusions

Bladder cancer is a major malignancy, causing a great percentage of cancer-related morbidity and mortality. Improving systemic treatment strategies for UC patients has been a challenge in recent years. Furthermore, due to its significant molecular and clinicopathologic heterogeneity, there is a huge clinical need for new promising therapeutic approaches. Ongoing efforts in bladder cancer have yielded significant improvements in the care of patients, especially in the metastatic setting, through *FGFR2/3* targeting, as well as with immunotherapy options with PD-L1 inhibitors. Furthermore, due to the molecular diversity in UC, there is vast potential for promising biomarkers in both neoadjuvant and metastatic settings. Although there are various hurdles to the advancement of precision medicine in UC, we are currently in the era of hope and promise with innovative clinical trials, multi-omics platforms, and increasingly refined methods for the future.

After decades of paucity in the therapies available, particularly in the metastatic setting, the advancement in genetics and increasing understanding of the biology of UC promise a future of truly efficacious treatments for our bladder cancer patients.

## Figures and Tables

**Figure 1 jpm-13-00756-f001:**
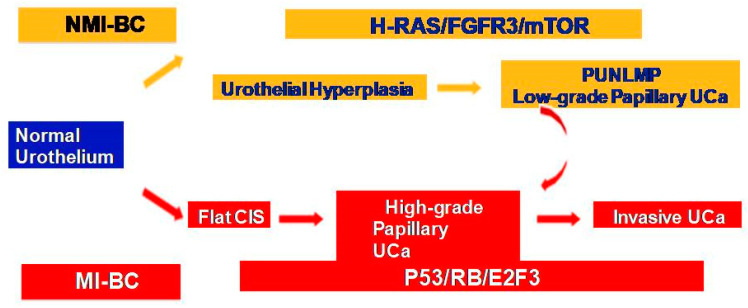
Two divergent molecular pathways in non-muscle-invasive and muscle-invasive UC.

**Figure 2 jpm-13-00756-f002:**
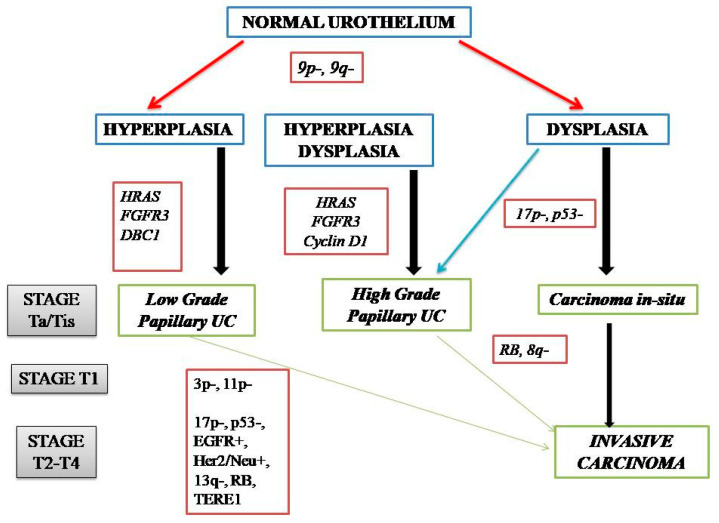
Genetic alterations in the pathogenic pathways of NMIBC and MIBC.

**Figure 3 jpm-13-00756-f003:**
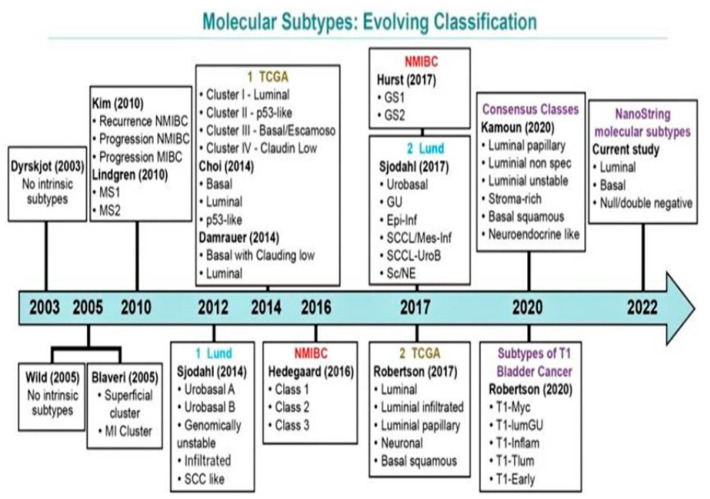
Evolving schemes of molecular classification of urothelial carcinoma of the bladder. Modified from Lopez-Beltran A, Cimadamore A, Montironi R, Cheng L. Molecular pathology of urothelial carcinoma. Hum Pathol 2021; 113:67–83 [26].

**Figure 4 jpm-13-00756-f004:**
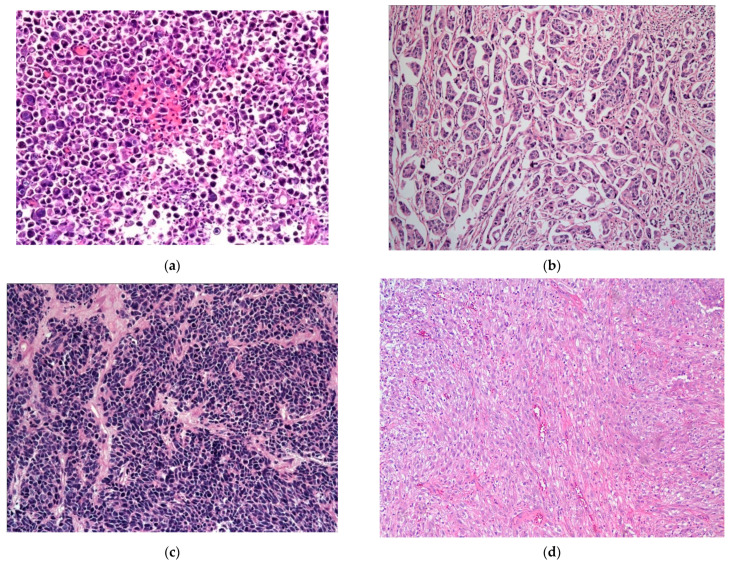
(**a**) Plasmacytoid subtype of UC. (**b**) Micropapillary subtype of UC. (**c**) Small-cell/neuroendocrine subtype of UC. (**d**) Sarcomatoid subtype of UC. (**e**) UC with divergent squamous differentiation. (**f**) UC with divergent glandular differentiation. (**g**) Nested subtype of UC.

**Figure 5 jpm-13-00756-f005:**
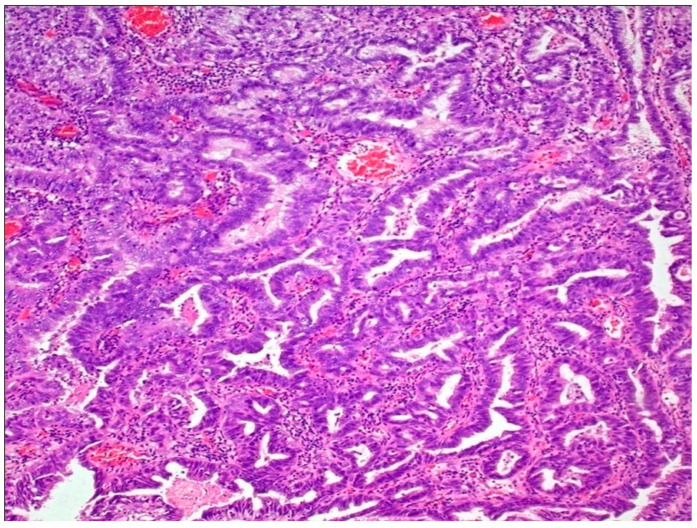
Adenocarcinoma of bladder.

**Table 3 jpm-13-00756-t003:** Targeted therapy approved in bladder cancer.

Molecular Target	Targeted Therapy	Clinical Trial [Ref.]	Patient Eligibility	Study Arms	Results	Comments
**FGFR**	**Erdafitinib**	BLC2001 [17,184]	Advanced UC and progression on prior platinum-based chemotherapy, with or without prior immunotherapy and with *FGFR* alterations (mutations/fusions)	Phase II single-arm study	ORR: 40% Median PFS: 5.5 months Median OS: 13.8 months Adverse events: stomatitis, hyponatremia, hyperphosphatemia	Accelerated FDA approval based on ORR. First gene-targeted therapy approved in UC.
** Nectin-4 ** (a cell adhesion molecule)	**Enfortumumab vedotin** (an antibody targeting Nectin-4 linked to a microtubule inhibitor conjugate (monomethyl auristatin E)	EV 201 [192]	Locally advanced or metastatic disease ineligible for cisplatin, not having received prior platinum-based chemotherapy, and previously treated with either a PD-1 or a PD-L1 inhibitor; no biomarker assay needed	Phase II single-arm study	ORR: 52% Adverse events: Neutropenia Rash Pneumonitis	Nectin-4 levels on tumor tissue are assessed with IHC. An H score is assigned with a range of 0–300, where 0 means no expression and 300 means maximal IHC staining.
		EV301 [187]	Locally advanced unresectable or metastatic UC (including those with squamous differentiation or mixed cell types) previously treated with platinum-based chemotherapy and PD-1/PD-L1 inhibitor; no biomarker assay needed	Enfortumab vedotin or investigator’s choice of chemotherapy (docetaxel, paclitaxel, or vinflunine)	Significant improvement in Median OS: 13 vs. 9 months, Median PFS: 6 vs. 4 months ORR: 41% vs. 18% Adverse events: Rash Peripheral neuropathy Hyperglycemia	FDA-approved for locally advanced or metastatic UC progressed on both platinum-based chemotherapy and immunotherapy.
** Trop-2 ** (a transmembrane glycoprotein highly expressed in most UC)	**Sacituzumab govitecan** (antibody–drug conjugate that targets Trop-2, and is coupled with SN-38, an active metabolite ofirinotecan)	TROPHY-U-01 [191]	Advanced UC previously treated with platinum-based chemotherapy or immunotherapy; no biomarker assay needed.	Single-arm phase II study	ORR: 27% Median PFS: 5 months Median OS: 11 months Adverse events: Neutropenia Anemia Thrombocytopenia	Advanced UC previously treated with platinum-based chemotherapy or immunotherapy.

**Table 4 jpm-13-00756-t004:** Other biomarkers for response to targeted therapy.

Biomarker	Study [Ref.]	Results	Comments
***AKT/PI3K/mTOR* and *MAPK* pathway **	**Bellmunt et al. [193]**	Responders to everolimus and *MTOR* inhibitor in mUC had mutations in *TSC1*, *TSC2*, and *mTOR.*	
***HRR* pathway**	**Grivas et al. [194]** **Powles et al. [195]**	In the phase II ATLAS study, rucaparib was not efficacious in mUC. Olaparib did not confer additional benefit when combined with durvalumab, even among those with *HRR* mutations.	
***HER* family genes**	**Necchi et al. [193]**	Responders to sorafenib in mUC demonstrated higher mutations in the *HER* family of genes, DDR genes, and *RAS/RAF* pathway.	

## Data Availability

Not applicable.

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
