# Peer review of "Precision Medicine in Bladder Cancer: Present Challenges and Future Directions"

_jpm, 2023, doi:10.3390/jpm13050756_

Round 1
Reviewer 1 Report
I congratulate the authors for this well-written review. The molecular classification of bladder cancer is an increasingly popular topic. The authors provide a comprehensive overview of precision medicine and its potential for personalized treatment based on tumor characteristics. This work offers valuable insights into future developments in this field.
Author Response
The changes have been highlighted in the revised manuscript in yellow.
Reviewer 2 Report
Dear Authors,
This study analyzes the current knowledge regarding the medical therapy of bladder cancer and its potential future developments and first of all, I congratulate with you for the topic that I found really interesting and very promising.
All I have to say is the paper is well organized, easy to read and scientifically correct.
Well done!
Author Response

(The authors gave the same response as above.)

Reviewer 3 Report
General comment
The manuscript entitled “Precision Medicine in Bladder Cancer: Present Challenges and Future Directions” represents a comprehensive summary of the current state of art of precision medicine, novel therapies and diagnostic pathways in bladder cancer. Despite the huge amount of information reported, the paper seems nevertheless messy in some sections and rushed in others. My main concern regards, therefore, the homogenization of paragraphs in order to avoid this effect along the text. I do not hide that reading some sections were interesting and enjoyable while others seemed to lack something, both in terms of content and readability. Suggested implementations are reported below in detail.
PS: Try to shorten the manuscript in order to avoid excessive redundancy with previously reported information
ABSTRACT
The abstract should be 200-250 words and, considering the great amount of data reported in the manuscript, it seems quite unsatisfiable compared to the text. I suggest you to revise the abstract in order to state what your manuscript reports and why it is important to the topic of your paper.
INTRODUCTION
I suggest the authors to briefly report the risk factors associated with BC, both environmental and genetic.
Conversely, the paragraph starting at line 75 could be shortened in favor of more information regarding the novelty of BC diagnosis and treatment.
MOLECULAR PATHOGENESIS
Check the quality of the figures.
MOLECULAR TAXONOMY OF UROTHELIAL CARCINOMA
The same comment could be applied to figure 3.
Shorten the section. It seems like an academic book on the topic. In addition, considering that the following paragraphs give more details on the molecular characterization of different subtypes, this section could be redundant with the data reported after.
I would suggest you also check the different headlines in order to give a clearer aspect of the manuscript.
MOLECULAR HETEROGENEITY AND SYSTEMIC TARGETED THERAPY
433: Too short, add recent studies and briefly discuss them.
441: Similarly to before. To this regard please also see: DOI: 10.3390/cancers14102545 and DOI: 10.3390/ijms23031133
445: A table reporting the currently ongoing clinical trials would be a nice addition.
PRECISION MEDICINE IN BLADDER CANCER
Probably, I would place this paragraph before the targeted therapies. However, this is your choice.
Briefly report the novel field of liquid biopsy in bladder cancer in terms of future perspective and state of art.
Avoid the use of too many bulleted lists.
700: avoid redundancy
Author Response

(The authors gave the same response as above.)
